# Merkel Cell Polyomavirus T Antigens Induce Merkel Cell-Like Differentiation in GLI1-Expressing Epithelial Cells

**DOI:** 10.3390/cancers12071989

**Published:** 2020-07-21

**Authors:** Thibault Kervarrec, Mahtab Samimi, Sonja Hesbacher, Patricia Berthon, Marion Wobser, Aurélie Sallot, Bhavishya Sarma, Sophie Schweinitzer, Théo Gandon, Christophe Destrieux, Côme Pasqualin, Serge Guyétant, Antoine Touzé, Roland Houben, David Schrama

**Affiliations:** 1Department of Pathology, Université de Tours, CHU de Tours, Avenue de la République, 37170 Chambray-les-Tours, France; serge.guyetant@univ-tours.fr; 2“Biologie des Infections à Polyomavirus” Team, UMR INRA ISP 1282, Université de Tours, 31 Avenue Monge, 37200 Tours, France; mahtab.samimi@univ-tours.fr (M.S.); patricia.berthon@inra.fr (P.B.); theo.gandon@etu.univ-tours.fr (T.G.); antoine.touze@univ-tours.fr (A.T.); 3Department of Dermatology, Venereology and Allergology, University Hospital Würzburg, Josef-Schneider-Straße 2, 97080 Würzburg, Germany; hesbacher_s@ukw.de (S.H.); wobser_m@ukw.de (M.W.); Sarma_B@ukw.de (B.S.); sophie.schweinitzer@gmx.de (S.S.); houben_r@ukw.de (R.H.); Schrama_d@ukw.de (D.S.); 4Dermatology Department, Université de Tours, CHU de Tours, Avenue de la République, 37170 Chambray-les-Tours, France; 5Plastic Surgery Department, Université de Tours, CHU de Tours, Avenue de la République, 37170 Chambray-les-Tours, France; aurelie.sallot@hotmail.fr; 6Neurosurgery Department, UMR 1253, i Brain, Université De Tours, CHU de Tours, Boulevard Tonnelé, 37044 Tours, France; christophe.destrieux@univ-tours.fr; 7CNRS ERL 7368, Signalisation et Transports Ioniques Membranaires, Equipe Transferts Ioniques et Rythmicité Cardiaque, Groupe Physiologie des Cellules Cardiaques et Vasculaires, Université de Tours, 31 Avenue Monge, 37200 Tours, France; come.pasqualin@univ-tours.fr

**Keywords:** Merkel cell carcinoma, histogenesis, polyomavirus, ATOH1, GLI1, sonic hedgehog, hair follicle

## Abstract

Merkel cell carcinoma (MCC) is an aggressive skin cancer frequently caused by the Merkel cell polyomavirus (MCPyV). It is still under discussion, in which cells viral integration and MCC development occurs. Recently, we demonstrated that a virus-positive MCC derived from a trichoblastoma, an epithelial neoplasia bearing Merkel cell (MC) differentiation potential. Accordingly, we hypothesized that MC progenitors may represent an origin of MCPyV-positive MCC. To sustain this hypothesis, phenotypic comparison of trichoblastomas and physiologic human MC progenitors was conducted revealing GLI family zinc finger 1 (GLI1), Keratin 17 (KRT 17), and SRY-box transcription factor 9 (SOX9) expressions in both subsets. Furthermore, GLI1 expression in keratinocytes induced transcription of the MC marker SOX2 supporting a role of GLI1 in human MC differentiation. To assess a possible contribution of the MCPyV T antigens (TA) to the development of an MC-like phenotype, human keratinocytes were transduced with TA. While this led only to induction of KRT8, an early MC marker, combined GLI1 and TA expression gave rise to a more advanced MC phenotype with SOX2, KRT8, and KRT20 expression. Finally, we demonstrated MCPyV-large T antigens’ capacity to inhibit the degradation of the MC master regulator Atonal bHLH transcription factor 1 (ATOH1). In conclusion, our report suggests that MCPyV TA contribute to the acquisition of an MC-like phenotype in epithelial cells.

## 1. Introduction

Merkel cell carcinoma (MCC) is an aggressive cutaneous neoplasm with a five-year overall survival rate of 40% [1]. Morphologically, MCC tumor cells display small cell carcinoma features and express both neuroendocrine and epithelial markers. In 2008, Feng et al. detected the sequence of a hitherto unknown polyomavirus integrated in the genomes of MCC tumor cells [2]. Subsequent studies revealed that approximately 80% of MCC cases are Merkel cell polyomavirus (MCPyV)-positive, and expression of the two viral T antigens (TA) (small T (sT) and large T antigens (LT)) are considered as the main drivers for carcinogenesis and growth of such tumors [2]. Interestingly, while several candidates, such as epithelial cells, fibroblasts, neuronal progenitors, or B cells, have been proposed, the nature of the cells giving rise to MCC following infection remains unknown [3,4,5,6].

Based on close phenotypic similarities, the eponymous Merkel cell (MC) was initially regarded as the most probable cell of origin of MCC. MCs can be found either in the appendages of the skin or in the basal layer of the epidermis. They function as mechanoreceptors capable of transmitting tactile stimuli onto Aβ-afferent nerve endings [7]. In mice and humans, MCs can be distinguished immunohistochemically from other intra-epidermal cells by positivity for the SRY-box transcription factor 2 (SOX2) and cytokeratins (KRT) 8, 18, and 20, which sequentially appear during MC differentiation and are also expressed by MCC [8,9,10,11,12].

For a long time, it was a matter of debate whether MCs develop from the neural crest or from the epidermal lineage [13]. Based on genetic mouse models, it is now widely accepted that MCs derive from epidermal progenitors in mammals [12,14,15] and that the transcription factor atonal homolog 1 (Atoh1) is the master regulator of this differentiation process [12,16,17]. While ectopic Atoh1 expression can induce MC differentiation throughout the epidermis of transgenic mice [3], physiological MC development preferentially occurs in hair follicles and in specialized structures named “touch domes” where the epithelial progenitors of MCs are located [18,19]. A critical step for MC differentiation in mice hairy skin is that these progenitors come into contact with dermal nerves leading to activation of the sonic hedgehog pathway (SHH) and subsequent GLI family zinc finger 1 (Gli1) expression [18,19]. Further markers characterizing these Gli1-expressing progenitors in mice are Krt17 [18], Sox9 [20], and CD200 [21], while only one study has shown KRT17 expression in human “touch dome” keratinocytes [22]. Notably, a high tumorigenic potential has been demonstrated for this cell population in transgenic models [23]. Therefore, these MCs’ epithelial progenitors, which remain poorly characterized in humans, are one potential candidate for MCC origin [24]. In contrast, due to lack of proliferative activity [25] and insensitiveness to oncogenic stimuli including ectopic TA expression [26], differentiated MCs are regarded as unlikely to be transformable [4].

Besides MCC, a second tumor entity known as trichoblastoma (TB) harbors cells with an MC phenotype. In this regard, TB as a benign epithelial skin tumor displaying hair follicle differentiation [27] is mainly composed of germinative basaloid cells, but is also characterized by sparse intra-tumoral MC cells. The latter probably reflects a preserved potential of TB cells to act as epithelial progenitors and, therefore, to differentiate into MCs [28,29,30]. Applying massive parallel sequencing on a combined tumor consisting of MCC and TB components, we recently demonstrated that MCPyV integration in a TB cell gave rise to an MCPyV-positive MCC [31] indicating that an MCPyV-positive MCC can arise from an epithelial cell. Moreover, the phenotypical similarities between TB and physiologic hair follicles, where MC progenitors are preferentially located, further support epithelial progenitors with intrinsic MC differentiation potential as possible ancestry for MCPyV-induced MCC [31]. In the present study, we first expanded characterization of such MC progenitors in humans and then aimed to evaluate how the viral T antigens might contribute to the development of an MC-like phenotype in this population using GLI1-expressing keratinocytes as a model system.

## 2. Results

### 2.1. MCs Are Often Located in Appendage Structures in Human Skin

MC development has mainly been characterized in mouse models [9,12,14]. Hence, in a first set of experiments, we used immunohistochemistry to compare the MC differentiation process under physiological conditions as well as in the tumor setting in humans. We started with characterizing the MC lineage by assessing physiological density and location of MCs in a set of 15 samples from three human autopsy skin specimens (Figure 1A,B, Appendix A, Appendix A). Mean MC density, regardless of the location, was 50 cells/mm^2^ of epidermis, and head and neck as well as acral skin were enriched in MCs compared to the other sites (density = 55 and 104 MCs/mm^2^, respectively). Moreover, MCs were often located in appendage structures (72% of all observed MCs), i.e., either hair follicles or sweat glands, as depicted in Figure 1B, Appendix A. Of note, contrary to previous reports, some dermal MCs were observed (Appendix A).

### 2.2. Cells with an MC Progenitor Phenotype Characterized by GLI1 Expression are Found in Close Proximity of MCs in Human Hairy Skin

Since MC epithelial progenitors can be expected to be found preferentially in regions enriched for MCs, we focused on the following on MC hotspots [18,22]. Such areas enriched in KRT20-positive MCs were mostly observed in hair follicles (52% of cases) or in junctions between eccrine sweat ducts and the overlying epidermis (36%). In the latter case, MCs were surrounded by clusters of verticalized basal keratinocytes resembling structures reported as “touch domes” [22] (Figure 1B, Appendix A, Appendix A). Slides of MC hotspots were subsequently stained for the epithelial progenitor markers GLI1, SOX9, and KRT17, revealing that epidermal cells surrounding MCs—in contrast to the rest of the epidermis—were characterized by nuclear GLI1 expression and positivity for the stem cell markers KRT17 and SOX9 (Figure 1B, Appendix A). In mice, Gli1-expressing keratinocytes in the hair follicle have been identified as MC progenitors [17,18,19,20]. Hence, our results demonstrate that also in human hairy skin an equivalent GLI1-positive population is preferentially located in the hair follicle.

### 2.3. GLI1 Expression in Keratinocytes Induces MC Lineage Markers

To evaluate a role of GLI1 expression in the establishment of the MC lineage in human epithelial cells, we used primary normal human epidermal keratinocytes (NHEK) as model system (Appendix A). These cells were transduced with a lentiviral vector encoding GLI1. Gene expression analysis after 14 days revealed an increase of the MC lineage markers *SOX2* (110-fold compared to the empty vector control, *p* = 0.002) and *KRT8* (4-fold, *p* = 0.05) in those cells (Figure 2A). Moreover, in GLI1-transduced cells *KRT17* and *SOX9* messenger RNA (mRNA) levels were found to be slightly elevated (2-fold), which, however, did not reach statistical significance. On protein level, we observed increased expression levels of SOX2 upon GLI1 expression by immunocytochemistry and immunoblot (Figure 2B, Appendix A). Additional immunostainings suggested enhanced KRT17 and SOX9 expression in GLI1-transduced NHEK, while no expression of the additional MC markers KRT8 or KRT20 was observed (Figure 2B, Appendix A). The discrepancy between induction of mRNA and lack of KRT8 protein in immunostaining upon GLI1 expression might be explained by protein levels below the detection limit of the antibody used. Nevertheless, together, these results suggest that GLI1, the executor of the sonic hedgehog pathway, is capable of initiating the first step of MC differentiation via SOX2 induction [6,9].

### 2.4. MC-Progenitor and MC Markers Are Expressed in Trichoblastoma and Merkel Cell Carcinoma

Next, we assessed how the markers defining the MC differentiation status are distributed in the two tumor entities harboring MC-like cells, i.e., TB and MCC. In five out of six MC containing interpretable TBs, we detected sparse SOX2-positive intra-tumoral cells. As typical for trichoblastoma, these expected “MCs” represented only a minority of cells dispersed within a vast majority of germinative tumor cells displaying a MC progenitor phenotype, and may be explained by germinative TB cells undergoing MC differentiation [30,32]. In line with this view and in line with the necessity of active hedgehog pathway signaling for potential MC differentiation in human epithelial cells [9,18], widespread nuclear GLI1 expression in the germinative cells was detectable in seven out of eight TB specimens (Table 1, Appendix A, Appendix A). Furthermore, diffuse expression of the GLI1 target genes, SOX9 and KRT17, was observed in germinative cells of all TB cases (Table 1, Appendix A, Appendix A). In conclusion, these results further substantiate known similarities between MCs’ epithelial progenitors and TB cells. In light of our previous report of an MCPyV-positive MCC arising from a TB cell [31], these observations further suggest such MC epithelial progenitors as a potential origin of MCPyV-induced MCC.

While in TB a mixture of cells with either epithelial progenitor or MC phenotype is present, almost all MCC tumor cells display a phenotype of mature MC. Indeed, in a previous study we observed 100, 99, and 92% of MCC cases with widespread positivity for the MC markers KRT8, 18, and 20, respectively [33,34]. Accordingly, in the present work, diffuse and strong nuclear positivity for SOX2 was detected in almost all analyzed MCC tumors (98%). While the MC progenitor marker KRT17 was not detectable (Table 1, Appendix A), GLI1 and SOX9 nuclear expression, representing the active forms of these transcription factors, were detected in 33% and 28% of cases, respectively (Table 1, Appendix A). Moreover, such findings were more frequently observed in MCPyV-negative than in MCPyV-positive cases (GLI1: 52 versus 24%, *p* < 0.03; SOX9 nuclear positivity: 81 versus 10%, *p* < 10^−9^, respectively) (Appendix A, Appendix A), suggesting that MCPyV presence is associated with a more mature MC phenotype.

### 2.5. T Antigens Can Trigger Early MC Differentiation Marker Expression in Epidermal Cells

On the supposition that MCC arises upon integration of MCPyV in a cell of the MC lineage, the virus might either hit an already determined MC cell or might trigger or promote the acquisition of the MC phenotype in an epithelial progenitor. To investigate a possible contribution of the MCPyV TAs to the development of an MC phenotype, sT and truncated LT were ectopically expressed in NHEK (Figure 3A). Notably, while cells could not be immortalized by the viral proteins, significant morphologic changes with reduction of cell size were observed upon TA expression (Figure 3A). Gene expression analysis after two weeks revealed an increase of mRNAs coding for early MC differentiation markers (*KRT8 p* = 0.02 and *KRT18 p* = 0.02), while the keratinocyte marker *KRT14* was slightly reduced upon TA expression (*p* = 0.09) (Figure 2B). Induction of KRT8 upon TA expression in NHEKs was confirmed by immunoblot and immunocytochemical staining, while no expression of SOX2 or KRT20 was observed in three independent experiments (Figure 3C,D, Appendix A). Interestingly, in situ KRT8 staining of TA-expressing NHEK demonstrated that expression of this marker was restricted to a subpopulation of cells with small-medium size and round shape (Figure 3C). 

### 2.6. T Antigens Induce Late MC Markers in GLI1-Expressing NHEK

To model TA expression in GLI1-expressing epithelial progenitor cells, we infected NHEKs with a bicistronic lentiviral construct coding for GLI1 and MCPyV-TA. After two weeks, morphological analysis of these cells in comparison to control cells infected with an empty vector revealed induction of a subpopulation of non-adherent, living cells forming clusters similar to the one observed for MCC cell lines (Figure 4A). Moreover, immunocytochemical staining revealed expression of the MC markers KRT8, SOX2, and, to a lesser extent, KRT20 (Figure 4B,C, Appendix A). Given that NHEKs represent only a limited model for MC progenitor cells, these findings—even though the detection of KRT20 was restricted to only a few cells—indicate that the interplay of GLI1 and MCPyV TA bears the potential of enforcing MC differentiation.

### 2.7. T Antigens Prevent ATOH1 Degradation

In NHEK, MCPyV-TA induced transcription of MC markers without significantly affecting *ATOH1* (Figure 3B), the known master regulator of MC differentiation [12,14]. Indeed, although LT-mediated ATOH1 induction was recently reported [35], we only observed a slight and statistically not significant mRNA increase upon TA expression. Hence, we hypothesized that the TAs might affect ATOH1 protein independent of gene transcription. To test this hypothesis, we transfected U2OS cells either with hemagglutinin (HA)-tagged ATOH1 alone or in combination with MCPyV-TA and analyzed RNA as well as protein levels. To this end, while the *ATOH1* mRNA level was not affected by TA co-expression, ATOH1 protein was increased (Figure 5A). Next, a constant amount of *ATOH1*-encoding plasmid (0.3 µg) and increasing amounts of TA-encoding plasmid (0–1.4 µg) were co-transfected, demonstrating a dose-dependent relation of increasing ATOH1 in the presence of MCPyV-TA (Figure 5B). Then, we asked whether this effect might be due to decreased protein degradation. To test whether protein stability is affected, the co-expression was performed while translation was inhibited in cycloheximide chase assays, allowing to assess ATOH1 protein decay in the presence or absence of TA (Figure 5C). These analyses revealed that TA increased ATOH1 half-life from 2 to 9 h. Interestingly, knockdown of TA expression in the MCC cell lines MKL-1 and WaGa failed to reduce ATOH1 protein levels (Appendix A) suggesting that in established MCC cells ATOH1 does not depend on stabilization by LT.

In mice, Atoh1 degradation has been shown to be controlled by phosphorylation of three carboxy-terminal serine residues (S331, S337, S341) leading to Atoh1 ubiquitinylation and subsequent targeting to the proteasome [36,37]. Hence, we speculated that TA-dependent stabilization might involve the respective sites in the human protein. Consequently, we generated expression constructs coding for ATOH1 proteins in which the serines were exchanged to alanines, either individually (S331A, S337A, and S342A) or all three combined (ATOH1-3A). Indeed, these modified ATOH1 proteins displayed extended half-lives in cycloheximide chase assays (Figure 5D, Appendix A). More importantly, however, while T antigens still stabilized ATOH1 proteins harboring single phospho-site mutations (Appendix A), no additional stabilization could be observed for the triple mutant protein (Figure 5D,E). Therefore, it is likely that the TAs act in the same pathway either by impacting phosphorylation of several serine residues on ATOH1 or by interfering with subsequent phosphorylation-dependent proteasome targeting.

### 2.8. The MCPyV Unique Region 1 (MUR1) in MCPyV LT Contributes to ATOH1 Stabilization

Irrespective of the fact that the exact mechanism of TA-mediated ATOH1 protein stabilization still requires further investigations, we finally wanted to know which of the two T antigens and which protein subdomains are involved in the process. Hence, we assessed ATOH1 protein levels after co-transfection of ATOH1 with either sT or LT, respectively. These experiments identified LT as the main effector of ATOH1 stability (Appendix A). To scrutinize which functional domain of large T might be involved in regulating ATOH1 degradation, another series of co-transfections was performed combining ATOH1 with LT mutants devoid of either specific interaction sites or the MCPyV unique region 1 (MUR1) region. Interestingly, mutants, which have been demonstrated to lack any growth-promoting activity, like the heat shock protein 70 (HSC70)-binding mutant D44N [38,39] or the RB transcriptional corepressor 1 (RB1)-binding deficient variants E216K and S220A [38], were still capable of mediating ATOH1 accumulation (Appendix A). However, co-transfection of ATOH1 with MCPyV-LT^ΔMUR1^, a LT variant still bearing growth-promoting activity [38], did not result in enhanced protein expression (Appendix A), suggesting that the MUR1 region of MCPyV-LT is essential for its ATOH1-stabilizing capacity. Since, however, the applied LT antibody (CM2B4) does not recognize LT^ΔMUR1^, we could not confirm that the protein was de facto expressed in these experiments (Appendix A). We, therefore, repeated this experiment with V5-tagged versions of LT and LT^ΔMUR1^. Now, both proteins were detectable and we again observed no stabilization of co-transfected ATOH1 in the case of LT lacking the MUR1 region (Appendix A). To further confirm the contribution of the MUR1 region, we also tested the truncated large T of AlDo, an MCC cell line expressing a truncated LT with an additional large deletion representing most of MUR1 [40]. Indeed, upon co-expression of AlDo LT, no stabilization but even a reduction of the ATOH1 protein level was observed (Appendix A).

## 3. Discussion

Today, the identification of the cell of origin for MCC is still pending. Based on the similarities in phenotype to MCs, the initially described “trabecular carcinoma of the skin” got its name MCC [24]. These phenotypic similarities can result either from transformation of the eponymous cell or inducing phenotypic changes during oncogenesis resulting in a phenotype resembling those cells. Since (1) MCs are regarded as post-mitotic cells with low sensitivity to oncogenic stimuli, (2) they demonstrate different preferred localizations compared to MCCs, (3) lack of infection of MCs by MCPyV, and (4) neuroendocrine tumors tend to derive from epithelial progenitor cells rather than end-differentiated cells [24], a direct transformation of MCs into MCCs is considered as quite unlikely. In this regard, we recently demonstrated that MCPyV integration in a TB gave rise to an MCPyV-positive MCC [31]. Of note, scattered MCs are frequently observed in TB [27,29,30], demonstrating that at least some of the cells possess the potential for MC differentiation, although the molecular determinants of this process are unknown.

Of note, the knowledge on MC development is mainly derived from mouse experiments. In the present study, we confirmed that MC hotspots are mostly located in the hair follicle in human hairy skin. In close vicinity to the MCs, we observed GLI1 and its downstream targets SOX9 [41] and KRT17 [42]-expressing keratinocytes. Similarly, we could confirm nuclear GLI1 positivity and related downstream SOX9 and KRT17 [27] expression in our TB cases suggesting that MC development under human physiological conditions as well as in TB tumors are quite similar and resemble the murine process with GLI1 activation being an early step. Accordingly, upon GLI1-expression in NHEK, we detected an increased expression of SOX9 and KRT17, and—as has been described for other cell lineages [43,44]—a prominent induction of SOX2. Since SOX2 can drive ATOH1 expression by binding to *ATOH1* enhancer [17] or *ATOH1* promoter [6] and thereby promote MC differentiation [45], SOX2 induction appears as a potential mechanism by which GLI1 promotes ATOH1-driven MC development. 

Based on our recently reported observation that a MCPyV-positive MCC could arise from a TB, we hypothesized that MCPyV oncoprotein expression is able to induce acquisition of a Merkel cell-like phenotype in epithelial progenitors with intrinsic MC differentiation potential. Indeed, while TA expression in NHEK reduced cell size, triggered KRT8 protein expression, and enhanced *KRT18* mRNA levels, we did not observe expression of KRT20, a marker appearing later during the MC differentiation process [9]. Although Atoh1 alone is able to initiate MC differentiation during embryonic mice development, Sox2 expression is required for Krt20 expression [9]. Accordingly, the two MCC tumors lacking SOX2 expression in our cohort were also KRT20 negative (data not shown). Hence, to test if the lack of KRT20 expression was due to a lack of SHH activation in NHEK, and subsequent lack of SOX2 expression, we generated a MC progenitor model system and assessed TA impact in it, by co-expressing GLI1 and TA in these cells. Although GLI1-expressing NHEKs represent only an artificial and limited model for MC progenitor cells, GLI1 and TA co-expression resulted in cells expressing SOX2 and KRT8, and even to a few cells displaying KRT20 positivity. Of note, similar as to what has been described for ectopic expression of LT in fibroblasts, we detected living cells with suspension growth. In contrast, however, we did not observe a different expression of Merkel cell markers between the adherent and floating cells (Appendix A).

While our results suggest SHH activation is required at some time point in MCC cell development, GLI1 expression was only observed in about 30% of cases in our study, which were mostly MCPyV-negative cases. Accordingly, therapeutic inhibition of SHH pathway using chemical inhibitors failed to reduce MCC tumor cell viability [46]. Therefore, SHH activation might contribute to MCC cell of origin establishment but then be lost during tumor development.

Another important factor in MC development is ATOH1. In this regard, induction of ATOH1 upon large T expression has been recently reported in fibroblasts [35]. In keratinocytes, we observed only a slight, statistically nonsignificant *ATOH1* mRNA level increase upon TA expression. This is in accordance with data obtained in mice where ectopic sT expression in combination with Atoh1 in epidermal cells did initiate a MC-like development [3], but only TA expression did not [3,47]. Thus, cellular context seems to influence the impact of LT expression on *ATOH1*. Indeed, we observed that ATOH1 degradation is impaired in the presence of LT in U2OS and 293 cells while TA knockdown does not affect ATOH1 protein levels in MCC cell lines. This might imply that TA only stabilizes ATOH1 in a specific environment. It is conceivable that, in a hit-and-run type mechanism (although the virus stays integrated in the host genome), LT contributes to initiating MC-like differentiation which later becomes independent of the viral protein. Indeed, T antigens are known to hijack many cellular processes [48], and stabilization of LT by sT via inhibition of the ubiquitin ligase “F-box and WD repeat domain containing 7” (SCF^Fbw7^) has been proposed [49], although this finding was recently called into question [50]. In mice, phosphorylation of the Atoh1 serine residues S328, S334, and S339 [36,37], equivalent to the amino acids S331, S337, and S342 in human, led to the ubiquitination of the protein by the ubiquitin ligase “HECT, UBA and WWE domain containing E3 ubiquitin protein ligase 1” (HUWE1) and subsequent targeting to the proteasome. Accordingly, human ATOH1 lacking the respective phosphorylation sites presented with an extended half-life in our study. Notably, while LT impaired degradation of wild-type ATOH1, it had no effect on mutant ATOH1. Hence, LT appears to affect the degradation process of ATOH1, either by interfering with the phosphorylation or ubiquitination step. With respect to the latter, although interactions between MCPyV-LT and SCF^Fbw7^ or “beta-transducin repeat-containing protein” (βTrCP) have been reported [49], these ubiquitin ligases appear as unlikely candidates since the (1) LT has been described as their target but not as targeting them, (2) sT, which inhibits both ubiquitin ligases, did not stabilize ATOH1, and (3) none of these ubiquitin ligases was shown to interact with ATOH1 [51]. In contrast, HUWE1 is a ubiquitin ligase that has been identified as ATOH1 binding partner using an unbiased comparative mass spectrometry approach [51]. Therefore, it is possible that the ubiquitin ligase HUWE1 is mediating the ATOH1 stabilization by MCPyV-LT. Moreover, our results suggest that for the ATOH1 stabilization MUR1 in LT is essential. In addition to the several unique functions of MCPyV-sT which have been described [52], this may contribute to the exceptional position of MCPyV among the polyomavirus family in being able to induce a neuroendocrine carcinoma of the skin. Furthermore, these observations suggest that the cell of origin of MCC might already display some degree of ATOH1 expression.

In the present study, we demonstrated that in a specific cellular context, i.e., GLI1-expressing keratinocytes, the expression of MCPyV T antigens can induce a MC-like differentiation. Moreover, the stabilization of ATOH1 by LT might enhance or promote the differentiation of the cell of origin toward an MCC phenotype.

## 4. Material and Methods

### 4.1. Human Samples

Healthy cutaneous tissues were obtained from dead people who had signed a body donation procedure for scientific purposes. Skin from five anatomic sites (scalp, face, trunk, finger, lower limb) were collected using a 6-mm-diameter punch in the 24 h following death, and then immediately fixed in formalin and then paraffin embedded. Fifteen TB cases were extracted from the archives of the Dermatology department of Würzburg (Local Würzburg Ethics Committee in Human Research, 196/12). After histological diagnosis confirmation by two pathologists (M.W., T.K.), only cases containing MCs were selected based on KRT20 immunostainings (*n* = 8). MCC cases enrolled in the present work were already included in a tissue microarray used in a previous study [34] (local ethics committee (Tours, France, N° ID RCB2009- A01056-51)). MCPyV status was previously determined using a validated real-time PCR [34].

### 4.2. Immunohistochemistry

Protein immunochemical detection was performed on formalin-fixed, paraffin-embedded (FFPE) samples (tissue), paraformalin-fixed (cytospin), or living cells. Immunohistochemical staining for KRT20, MCPyV-LT, Neurofilament, and SOX9 were performed using a BenchMark XT Platform, as instructed [34,53]. Immunohistochemical staining for GLI1, KRT8, KRT17, KRT18, and SOX2 as well as all cytospin stainings were performed manually. Microscopic evaluation was performed by a pathologist (T.K.). All details regarding antibodies and dilutions are provided in Appendix A.

### 4.3. Samples’ Management and Interpretation of Immunohistochemical Staining

To determine MC densities, 250 consecutives 5-μm-thick sections were cut from FFPE healthy cutaneous tissues (6-mm-diameter skin punches cut into two equal parts). Every 10th slide, a KRT20 immunohistochemical staining allowing the detection of MC was performed, i.e., one KRT20-stained slide every 50 μm. Unstained slides were preserved for further analyses (MC progenitor markers’ evaluation). MC number and location (interfollicular epidermis, hair follicle (infundibulum or isthmus), sebaceous, or sweat glands) were then assessed by a pathologist (T.K.). Since MC are frequently located in the connection area between epidermis and an appendage, i.e., either hair follicles or sweat glands, all MCs located in front of an appendage structure (hair follicle, ostium of a sweat gland or sweat gland duct) were considered to belong to this appendage. Of note, MC hotspots were defined as areas with more than three MCs in one microscopic field at high magnification. Densities of MCs and related hotspots were estimated, taking cut thickness and length of the skin sample into account (estimated evaluated surface = 14.74 mm^2^/punch). Unstained slides adjacent to the hotspots were consequently investigated for MC progenitor markers.

### 4.4. Primary Keratinocytes and Cell Lines

After informed written consent of the patients (*n* = 3), normal human epidermal keratinocytes (NHEK) were extracted, respectively, from abdominal human samples obtained from the plastic surgery of the University Hospital center of Tours (France) using previously described protocols [40,54,55,56] (Local Ethics Committee in Human Research, Tours, France; no. ID RCB2009-A01056-512016 064). NHEK were cultured in Keratinocyte Serum-Free Medium (K-SFM; Invitrogen Life Technologies), supplemented with epidermal growth factor (5 ng/mL) and bovine pituitary extract (50 μg/mL; all purchased from Invitrogen Life Technologies) at 37 °C, 5% CO_2_ in a humidified incubator. HEK293 (RRID: CVCL_0045), HEK293T (RRID: CVCL_0063), U2OS (RRID: CVCL_0045), the MCC cell line WaGa (RRID:CVCL_E998), and MKL-1 (RRID:CVCL_2600) [57] were cultivated in Roswell Park Memorial Institute (RPMI) 1640 supplemented with 10% Foetal calf serum (FCS), 100 U/mL penicillin and 0.1 mg/mL streptomycin. HEK293 (RRID: CVCL_0045) and U2OS (RRID:CVCL_0045) were used for co-transfection experiments. HEK293T (RRID:CVCL_0063), i.e., HEK293 expressing SV40 T antigens, were used for lentivirus production. The MCC cell line WaGa was included as positive control for immunostaining of MC markers.

### 4.5. Lentiviral Vectors’ Generation and Transduction Protocol

The pFLAG-CMV-4-GLI1 plasmid was kindly provided by Dr. J. Vachtenheim (Czech Republic) [58]. *GLI1* was subcloned into pFLAG-CMV backbone (System Biosciences) containing puromycin resistance by classical cloning. Phosphosite mutations (S331A, S337A, S341A) were introduced in *ATOH1* sequence using the Quickchange Lightning mutagenesis kit (Agilent, Frankfurt, Germany) [59]. All TA- and LT-expressing pCDH vectors were previously described [38]. GLI-IRES-TA sequence was cloned into a pCDH backbone. For inducible knockdown of MCPyV-LT, we used the lentiviral single vector TA.shRNA.tet, allowing constitutive green fluorescent protein (GFP) expression and doxycycline (Dox)- inducible expression of an shRNA targeting all transcripts derived from the MCPyV early region [59]. Lentiviral supernatants were produced in HEK293T cells as previously described [60]. Harvested virus supernatant was sterile filtered (0.45 μm) and polybrene (1 μg/mL) was added for infection. Lentiviral transduction of NHEK was performed after seven days of culture. Then, 14–20 h after infection, target cells were washed with medium. NHEK were then subjected to antibiotic selection (puromycin). NHEK were analyzed two weeks after transduction.

### 4.6. Gene Expression Analyses

Total cellular RNA was isolated by using the peqGOLD total RNA kit (VWR; Darmstadt, Germany) with a subsequent DNaseI digestion step according to the manufacturer’s instructions. For cDNA synthesis, the Superscript II RT First Strand Kit (Invitrogen GmbH, Karlsruhe) was used. PCR primer sequences used to detect *ATOH1*, *GLI1*, *KRT8*, *14*, *17*, *18*, *20*, *RPLP0*, *SOX2,* and *SOX9* are given in Appendix A. Thermal profile for the PCR using the Takyon Low Rox Sybr MasterMix (Eurogentec; Cologne, Germany) contained an initial denaturation step at 95 °C for 10 min, followed by 40 cycles of two-step PCR including 15 sec at 95 °C and 60 sec at 60 °C. Quantification was performed in three independent experiments.

### 4.7. Immunoblot

Cells were lysed in 0.6% SDS, 1 mM Ethylenediaminetetraacetic acid (EDTA), 10 mM Tris- HCl (pH 8.0), 2 mM NaF, 2 mM NaVO3 supplemented with a protease inhibitor cocktail (Roche Diagnostics, Basel, Switzerland). Samples were resolved by SDS-PAGE, transferred to nitrocellulose membrane, blocked for 1 h with Phosphate buffered saline (PBS) containing 0.05% Tween 20 and 5% powdered skim milk, then incubated overnight with anti-HA (ab18181, Abcam, 1:1000), LT (CM2B4, Santa Cruz, 1:200), sT (2T2, Hybridoma obtained from C. Buck laboratory), anti-GLI1 (C68H3, Ozyme, 1:200), anti-SOX2 (EPR3131, Abcam, 1:200), anti-ATOH1 (polyclonal, Proteintech, 1:600), or anti-Actin antibody (A5441, Sigma, 1:1000), washed three times with PBS with 0.05% Tween 20 (PBS/Tween), then incubated for 1 h with a peroxidase-conjugated secondary antibody. Finally, following three washes with PBS/Tween, respective proteins were detected by using a chemiluminescence detection procedure. All primary Western blot membranes’ acquisition without cropping and intensity adjustment are available in Appendix A.

### 4.8. Transient Transfection and ATOH1 Half-Life Evaluation

Transient transfections were done using 2 μg of DNA with polyethylenimine (PEI) and protein expression was analyzed 24 h after transfection. For ATOH1 half-life determination, 24 h after transfection, cells were exposed to cycloheximide (0.3 mg/mL) in a time-course experiment. After harvesting, protein expression was then investigated by immunoblotting, and quantification was performed using ImageJ software.

### 4.9. Flow Cytometry

Anti-CD200 phycoerythrin (PE)-conjugated (OX-104, BioLegend) and anti-leucine rich repeat containing G protein-coupled receptor 6 (LGR6) Allophycocyanin (APC)-conjugated (Sc-393010, SantaCruz) antibodies were used for NHEK characterization.

### 4.10. Image Analysis and Expression Score Determination

Cell morphology was analyzed on adherent living cells. After acquisition of five adjacent microscopic fields, cell contouring was performed on 100 cells per conditions (three independent experiments) and cell size was then analyzed using ImageJ software. For protein expression evaluation, 2 × 10^5^ cells were fixed in formalin, spotted on slides, and submitted to immunohistochemical staining. Stained slides were scanned by using NanoZoomer (Hamamatsu, Hamamatsu City, Japan). Computation of the expression score after transduction was performed with a custom software written in ImageJ Macro language. Briefly, color range for each staining was first defined from the whole image data set. Afterwards, cells were segmented in each image. For each cell-related area, the percentage of each type of viral protein staining (low, medium, and high) was computed. H-score was finally calculated for each cell with the following formula:$$Hscore=\frac{\left(lowstainingarea\times 1\right)+\left(mediumstainingarea\times 2\right)+\left(highstainingarea\times 3\right)}{totalcellarea}$$

Analysis was initially performed on 10 consecutive fields (magnification × 10). In cases in which fewer than 1000 cells per conditions were analyzed, new acquisitions were performed in order to reach this minimal limit of analyzed cells. Results were subsequently expressed as median, quartiles Q1–Q3, and 1st–99th percentiles of the complete cell population analyzed. Protein quantification on immunoblot was performed by ImageJ using the “gel analysis” function.

### 4.11. Statistical Analysis

Continuous data are described as mean with standard error of mean (SEM), and categorical data with number and as percent. Associations were assessed by two-tailed Fisher exact test for categorical data and Mann–Whitney test for continuous data. Paired *t* test was used for RNA expression analysis without multiple testing correction. The *p* < 0.05 was considered statistically significant. XL-Stat-Life (Addinsoft, Paris, France) was used for statistical analyses.

## 5. Conclusions

Whether MCC is derived from MC or from another skin lineage is a long-time matter of debate. In this regard, we recently demonstrated that MCPyV integration in a TB gave rise to an MCPyV-positive MCC [31] and, consequently, postulated that MCC tumorgenesis can be initiated in MC epithelial progenitors. In the present work, we confirmed the close similarities between TB tumor cells and epithelial MC progenitors, evident by expression of GLI1 and its related downstream targets, i.e., KRT17 and SOX9, in both settings. While a mixture of cells with either MC progenitor phenotype or already differentiated MCs was observed in TB, almost all MCC tumor cells display a fully differentiated MC phenotype. Consequently, we assessed if TA could contribute to the acquisition of an MC phenotype. In accordance with this hypothesis, ectopic TA expression in NHEK led to induction of early MC markers while concomitant induction of SOX2, KRT8, and KRT20 were only achieved upon co-expression of TA and GLI1. Therefore, our results suggest that TA can induce acquisition of Merkel cell-like phenotype when expressed in epithelial MC progenitors. Accordingly, since large T antigen extends ATOH1 half-life, ATOH1 stabilization by MCPyV oncoproteins might further contribute to the MC-like phenotype observed in MCC.

## Figures and Tables

**Figure 1 cancers-12-01989-f001:**
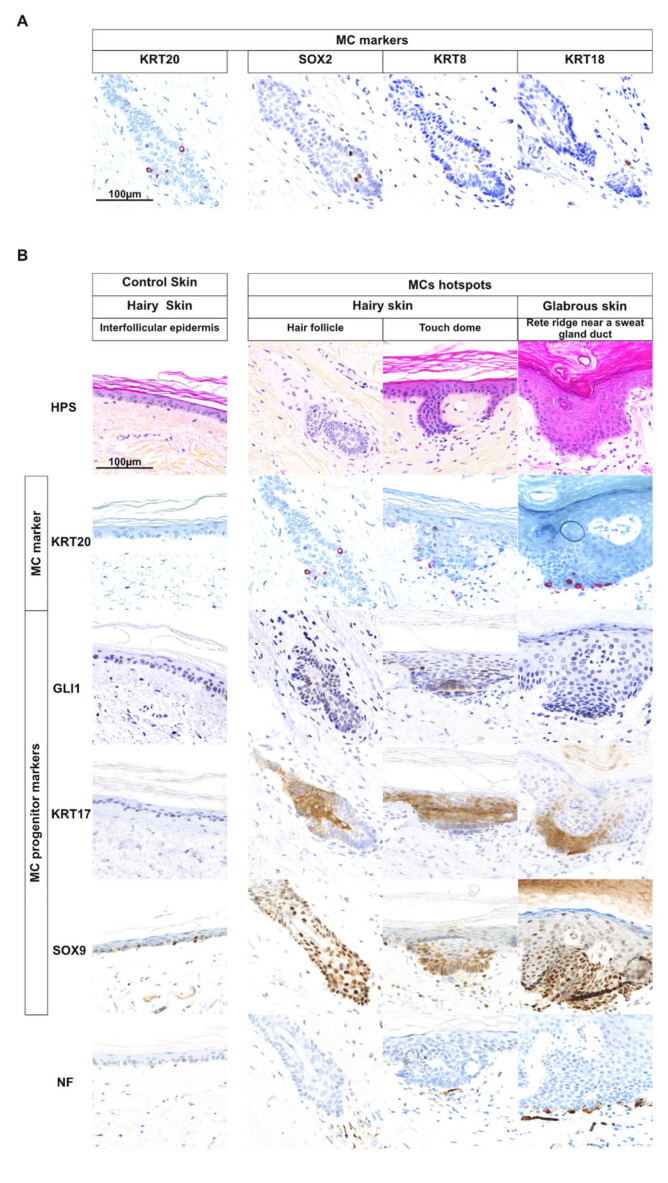
Merkel cells and possible Merkel cell progenitors in human skin. (**A**) Keratin 20 (KRT20), SRY-box transcription factor 2 (SOX2), KRT8, and KRT18 staining was used to identify Merkel cells (MCs) (bar = 100 µm) (only one hotspot investigated for illustration purpose). Merged analysis is available in Appendix A. (**B**) Identification of potential MC progenitors in human skin: Three MC hotspots as well as interfollicular epidermis for comparison are depicted (bar = 100 µm) (15 hotspots investigated in total). Immunohistochemical staining revealed expression of KRT17 and SOX9 in the epidermal cells surrounding differentiated MCs suggesting that these cells are MC progenitors. Nuclear GLI family zinc finger 1 (GLI1) was detected only close to MC hotspots in hairy, but not in acral skin. Of note, neurofilament (NF)-expressing dermal nerves were observed in contact with the MCs.

**Figure 2 cancers-12-01989-f002:**
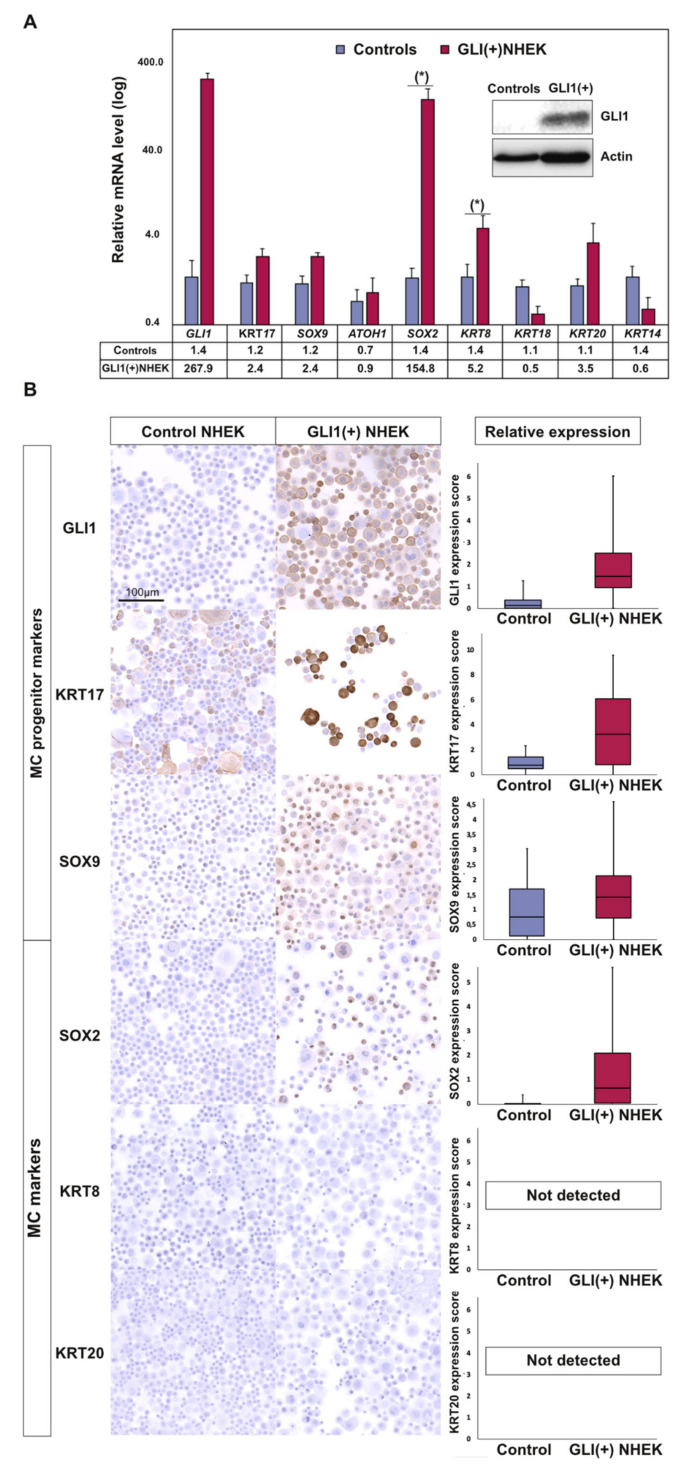
Ectopic GLI1 expression in primary human epidermal keratinocytes induces several MC lineage markers: Normal human epidermal keratinocytes (NHEK) were infected with a lentiviral vector coding for GLI1 and puromycin resistance. Following antibiotic selection, cells were harvested after 14 days of cultivation. (**A**) Immunoblot analysis was performed to confirm GLI1 expression (insert), and isolated RNA was subjected to complementary DNA (cDNA) synthesis and real-time PCR. Relative messenger RNA (mRNA) expression levels of the indicated Merkel cell lineage markers are given as mean (+ standard error of the mean (SEM)) of four independent experiments (* *p* value < 0.05, paired *t* test) (mean CT value of the controls was used as reference). (**B**) Expression of GLI1, the MC progenitor (KRT17, SOX9) and the MC markers (SOX2, KRT8, and KRT20) was assessed by immunohistochemistry and relative protein expression quantification was performed on at least 1000 cells/condition using ImageJ software. Results are displayed as box and whiskers diagram with median, Q1, and Q3, as well as first and 99th percentile. These results were confirmed by two additional independent experiments (immunostaining and immunoblot) as shown in Appendix A. Uncropped membranes and Western blot signal quantifications are available in Appendix A, respectively.

**Figure 3 cancers-12-01989-f003:**
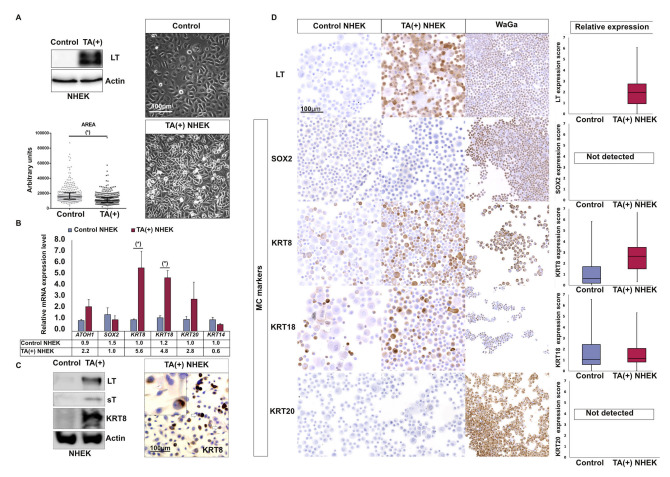
T antigens induce expression of some early MC differentiation markers in primary human keratinocytes. **A:** NHEKs were infected with a lentiviral vector coding for small T (sT) and truncated Large T (LT) as well as a puromycin resistance. Following antibiotic selection, cells were analyzed after 14 days of cultivation. (**A**) Immunoblot analysis confirmed LT expression, and microscopic inspection revealed a less-flattened phenotype and cultures reaching much higher densities. Under microscopic examination such cells harbored reduced cytoplasmic size compared to the controls, as confirmed using imageJ software (bar = 100 µm) (* *p* value < 0.05, Mann–Whitney U test, *n* = 3 independent experiments). (**B**) Relative mRNA levels of the indicated Merkel cell differentiation markers (* *p* value < 0.05, paired *t* test, *n* = 4 independent experiments), (**C**) Immunoblot demonstrated T antigens (TA)-induced KRT8 protein expression and immunohistochemistry additionally revealed KRT8 expression is restricted to a subpopulation of small- to medium-sized round cells. Furthermore, occasionally “dot like” staining was observed (white arrows). (**D**) Immunohistochemical assessment of the indicated MC markers in TA-expressing NHEK, control NHEK and the MCC cell line WaGa (bar = 100 µm). KRT8 induction by T antigens was confirmed in two additional independent experiments, which are depicted in Appendix A. For relative quantification of protein expression levels, at least 1000 cells/condition were evaluated using ImageJ software. Results are displayed as box and whiskers diagram with median, Q1, and Q3 as well as first and 99th percentile. Uncropped membranes and Western blot signal quantifications are available in Appendix A, respectively.

**Figure 4 cancers-12-01989-f004:**
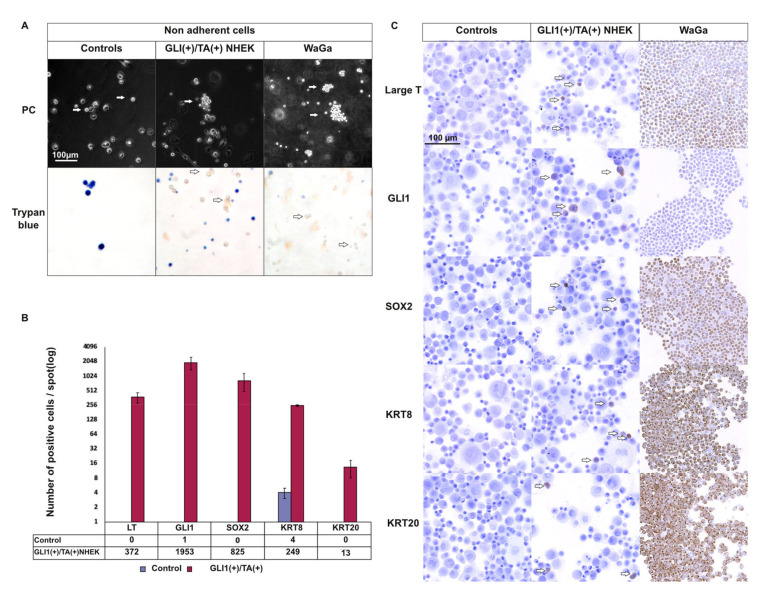
Induction of late MC markers by combined expression of GLI1 and Merkel cell Polyomavirus (MCPyV) T antigens (TA) in primary keratinocytes. NHEKs were infected with a bicistronic lentiviral vector coding for GLI1 as well as sT and truncated LT. Under control of a second promoter, a pure resistance was expressed. Following antibiotic selection, cells were analyzed after 14 days of cultivation. (**A**) GLI1/TA combined ectopic expression was associated with formation of floating clusters of living cells in normal human epidermal keratinocytes (NHEK), while these findings were not observed in controls or when GLI1 and TA were transduced independently (PC: Phase contrast) (Appendix A) (*n* = 3 independent experiments). White arrows indicate the floating cells. (**B**,**C**) Immunohistochemical assessment of Merkel cell markers (SOX2, KRT8, and KRT20) expression levels in GLI1/T antigen-expressing NHEKs and controls. Immunohistochemistry was performed on the respective cells spotted on slides (2 × 10^5^ cells/condition). B. Count of cells expressing the Merkel cell markers in GLI1/T antigens (TA)-expressing NHEK and controls (results are mean ±SEM of three independent experiments). Counting of positive cells was preferred to relative protein level quantification due to the low number of GLI1/TA-expressing cells. C. Representative photos of LT, GLI1, SOX2, KRT8, KRT18, and KRT20 expression in NHEK (controls), GLI1/TA-expressing NHEK, and the WaGa MCC cell line. White arrows indicate cells expressing the respective proteins. The results for two additional independent experiments are shown in Appendix A.

**Figure 5 cancers-12-01989-f005:**
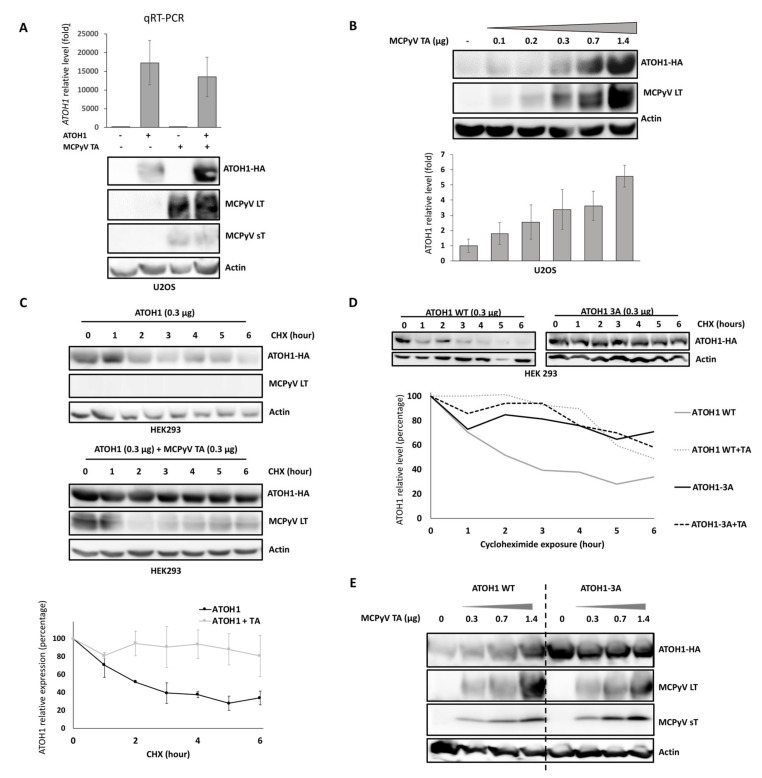
MCPyV T antigens increase the half-life of ATOH1 (**A**) **Hemagglutinin** (HA)-tagged ATOH1- and/or TA-encoding plasmids were transfected either individually or combined into U2OS cells. After two days, real-time PCR and immunoblot analyses were performed. While *ATOH1* mRNA was not affected (mean ± SEM of three independent experiments), ATOH1 protein accumulation in the presence of TA was observed. (**B**) Co-transfection of a constant amount (0.3 µg) of HA-tagged ATOH1 and increasing amounts of TA in U2OS cells followed by immunoblot analysis. ATOH1-HA signals relative to actin were quantified using ImageJ. Mean ± SEM of three independent experiments was displayed. (**C**) Evaluation of ATOH1 half-life in absence or presence of T antigens. Twenty-four hours after transfection, HEK293 cells were exposed to the translation inhibitor cycloheximide (CHX) for variable durations (0–6 h). ATOH1-HA expression was then evaluated by immunoblot analysis and quantified using the Image J Software (mean ± SEM of three independent experiments are depicted). (**D**) A mutant of ATOH1-HA, in which the three serines at positions 331, 337, and 342 were all exchanged to alanines (ATOH1-HA-3A), was generated, and the impact of co-transfected TA on ATOH1-HA wild type and ATOH1-HA-3A expression was analyzed in CHX chase experiments (see C). Quantified signals relative to actin are given in the graphs below. (**E**) Co-transfection of ATOH1-HA-3A with increasing amounts of TA did not affect ATOH1 protein expression level (this was confirmed in a second independent experiment). Uncropped membranes and Western blot signal quantifications are available in Appendix A, respectively.

**Table 1 cancers-12-01989-t001:** Expression of Merkel cell progenitor markers in trichoblastoma (*n* = 8) and Merkel cell carcinoma (*n* = 103).

MC Progenitor Markers	TB (*n* = 8 Cases)	MCC (*n* = 103 Cases)
**GLI1**		
Negative	1 (13%)	60 (67%)
Positive (nuclear)	7 (87%)	29 (33%)
No data available	0	14
**KRT17**		
Negative	0	94 (100%)
Positive (cytoplasmic)	8 (100%)	0
No data available	0	9
**SOX9**		
Negative	0	7 (8%)
Dot-like (cytoplasmic)	0	59 (64%)
Patchy (nuclear)	0	26 (28%)
Diffuse (nuclear)	8 (100%)	0
No data available	0	11
**MC markers**	**TB**	**MCC**
**SOX2**		
Negative	1 (17%)	2 (2%)
Positive (nuclear)	5 (83%)	94 (98%)
No data available	2	7
**KRT20**		
Negative	0	8
Diffuse (cytoplasmic)	8 (100%)	2
Mixed (cytoplasmic)	0	66
Dot-like pattern (cytoplasmic)	0	19
No data available	0	8

KRT: Cytokeratin; GLI1: GLI family zinc finger 1; MC: Merkel cell; MCC: Merkel cell carcinoma; SOX2: SRY-box transcription factor 2; SOX9: SRY-box 9, TB: Trichoblastoma. Representative photos of SOX9 expression patterns are available in Appendix A. Results are given as numbers and percentage of interpretable cases.

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
