# Peer review of "Merkel Cell Polyomavirus T Antigens Induce Merkel Cell-Like Differentiation in GLI1-Expressing Epithelial Cells"

_cancers, 2020, doi:10.3390/cancers12071989_

Round 1
Reviewer 1 Report
The manuscript has been significantly improved in this revised version. The authors have sufficiently addressed my concerns raised during the last review of this manuscript. I support its publication in Cancers.
Reviewer 2 Report
The revised manuscript does address the concerns raised and can now be accepted in the present form.
This manuscript is a resubmission of an earlier submission. The following is a list of the peer review reports and author responses from that submission.
Round 1
Reviewer 1 Report
Merkel cell carcinoma is a highly aggressive skin cancer. At least 60% of all MCC is caused by integration of Merkel cell polyomavirus (MCPyV) with continued expression of truncated large T antigen (LT) and small T antigen (sT). An important question in the field is the cell of origin particularly for MCC tumors containing MCPyV. A variety of studies in mice have implicated the ATOH1 transcription factor and Hedgehog signaling and the downstream GLI1 transcription factor in Merkel cell development.
Of note, in a previously published work, this group had demonstrated that a MCPyV-containing MCC tumor developed from a trichoblastoma (TB), a benign skin tumor with features of hair follicle differentiation that can also have an increased number of Merkel cells. In this manuscript, the authors compare expression of several differentiation markers in normal human (cadaveric) skin, MCC tumors, trichoblastoma neoplasms, normal human embryonic keratinocytes (NHEK), and the osteosarcoma U2OS cell line. They also test the impact of T antigen expression and GLI1 expression in NHEK cells and the effect of LT on the levels of ATOH1 in U2OS cells.
The authors propose that MCPyV T antigens can induce Merkel cell-like differentiation in skin epithelial progenitor cells. However, it is not clear if they have done enough to justify that claim. Instead, they compare specific IHC markers in normal skin as well as trichoblastoma and MCC tumors and observed that expression of several markers overlapped. Then they expressed GLI1 and MCPyV in NHEK cells and observed changes in expression of several markers including SOX2 and SOX9 and could induce the cells to grow as an adherent culture to suspension. In addition, they also observed that MCPyV LT can increase the half-like of the ATOH1 transcription factor. So while the authors have some evidence that MCPyV LT/sT alone, and in combination with GLI1, can induce changes in specific gene expression and cellular phenotype they have not introduced T antigens in a skin progenitor cell.
In Figure 1, they examine IHC markers associated with Merkel cells (MC) in normal skin. They observed that MCs are found in association with hair follicles and touch domes in hairy skin as well as in rete ridge near sweat gland in glabrous skin. They observed that MC found in hairy skin stain with SOX2, KRT8, KRT18, KRT20, and neurofilament. In contrast, GLI1, KRT17, SOX9 stain cells in close association with MC.
In Table 1, they compare trichoblastoma with MCC by staining for several IHC markers. They observed that GLI1 was positive in most trichoblastomas (7/8) and in many (29/89) MCC. Similarly, SOX2, SOX9 and KRT20 was present in most trichoblastomas and MCC tumors. In contrast, KRT17 was detected in all TB and no MCC. It is reasonable to conclude that the epithelial trichoblastoma tumor has features similar to MCC. However, it is not clear that a trichoblastoma has any overlap with a skin progenitor cell.
In Figures 2, 3, and 4, the authors overexpress GL1 and MCPyV T antigens in NHEK. They observed that ectopic GLI1 expression in NHEK cells led to significantly increased levels of SOX2 and KRT8 and slightly increased levels of KRT17, SOX9, and KRT20. In contrast, expression of sT and LT led to significantly increased levels of KRT8 and KRT18 but not SOX2 or KRT20. Combined expression of GLI1 with sT and truncated LT led to increased levels of SOX2, KRT8, and KRT20 in double positive GLI1+/TA+ cells. In Supplementary Figure 4c and 4d, it appears that in the adherent and suspension cells, not all cells express LT, GLI1, and SOX2. It would be useful to quantify the relative fraction of adherent versus suspension cells that express LT, SOX2, and GLI1 and the relative fraction of adherent and suspension cells.
In Figure 5, they observed that LT could stabilize levels and increase the t½ of ATOH1 in U2OS cells. This effect of LT was dependent on three serine residues in ATOH1 (S331, S337, S342) as well as the MCPyV unique region 1 (MUR1, residues 101-208) in LT. However, there was no additional experiments performed to understand how LT could affect ATOH1 stability. The MUR1 domain has been implicated in recruiting the ubiquitin ligases Fbw7 and beta-TrCP. Does the LT affect on ATOH1 stability involve either Fbw7 or beta-TrCP? Does ATOH1 show similar stability in MCPyV containing MCC cell lines? Does the Aldo cell line containing a LT missing most of the MUR1 domain also stabilize ATOH1?
Minor
Figure 3, panel C. “… immunhistochemistry additionally reveals KRT8 expression is restricted to a subpopulation of small to medium-sized round cells. Furthermore, occasionally “dot like” staining was observed (white arrows).” The magnification or clarity of the image does not allow the reader to come to the same conclusion.
Reviewer 2 Report
In this manuscript, Kervarrec and colleagues describe their characterization of Merkel cell (MC) progenitor cells in humans and the role of MCPyV T antigens (TAs) in the development of an MC-like differentiation phenotype in skin epithelial progenitor cells. Recently, through genetic analysis of a very rare MCPyV-positive Merkel cell carcinoma (MCC) within trichoblastoma (TB), a benign epithelial follicular tumor, the authors found that the integration of MCPyV genome in TB cells could potentially give rise to an MCPyV-positive MCC. Based on this finding and other phenotypic analysis, the authors propose that MCPyV-positive MCC may evolve from epithelial cells such as Merkel Cell (MC) progenitor cells. In this study, they detected the expression of GLI1, KRT17, and SOX9 in both TB and human MC progenitors. They found that GLI1 expression in keratinocytes could induce MC marker SOX2, supporting its role in human MC differentiation. They also found that the expression of MCPyV TAs in human keratinocytes led to the induction of an early MC marker KRT8, while co-expression of GLI1 and TAs stimulated a more advanced MC phenotype expressing late MC markers such as SOX2, KRT8, and KRT20. Finally, they demonstrated that MCPyV-Large T Antigen can inhibit the degradation of the MC master regulator ATOH.
These important findings shed new light on the cell of origin for MCPyV-positive MCC and suggested that MCPyV TAs can promote the development of an MC-like phenotype in epithelial progenitor cells.
The manuscript could be further strengthened if the following concerns could be addressed:
The western blot in Fig. 2A does not show equal loading. It appears that a lot more Actin is present in the GLI1(+) lane.
Fig. 1D was mentioned several times in the manuscript but it is not found in any part of the manuscript.
The IHC data shown in Fig 2B is not consistent with the mRNA data shown in Fig. 2A.
Is sT expressed in the cells of Figs. 3-5 experiments?
NHEK cells were used for most of the experiments presented in this study. Not sure why the authors switched to U2OS cells for Figure 5. It will make sense to repeat this experiment in NHEKs.
The sample size used in this study is not clear. For example, in Table 1. it is unclear what n stands for. A total number of cells or tissues? It is also not clear how many tissues were stained to obtain the results shown in each figure.
Reviewer 3 Report
I enjoyed reading authors take on the origin of Merkel cell-like 2 differentiation in skin epithelial progenitor cells mediated by MCPyV T antigens. The results were backed by string evidence.
Author Response
We thank reviewer 3 for this very positive feedback regarding our work.